# The *Drosophila drop-dead* gene is required for eggshell integrity

Tayler D. Sheahan[¤a], Amanpreet Grewal, Laura E. Korthauer[¤b], Edward M. Blumenthal *

Department of Biological Sciences, Marquette University, Milwaukee, Wisconsin, United States of America

¤a Current address: Pittsburgh Center for Pain Research and Department of Neurobiology, University of Pittsburgh, Pittsburgh, Pennsylvania, United States of America
¤b Current address: Department of Psychiatry and Human Behavior, Alpert Medical School of Brown University, Providence, Rhode Island, United States of America
* edward.blumenthal@marquette.edu

**Data Availability Statement:** All relevant data are within the paper and its Supporting Information files or are held in a public repository at https://epublications.marquette.edu/sheahan_2023/

## Abstract

The eggshell of the fruit fly *Drosophila melanogaster* is a useful model for understanding the synthesis of a complex extracellular matrix. The eggshell is synthesized during mid-to-late oogenesis by the somatic follicle cells that surround the developing oocyte. We previously reported that female flies mutant for the gene *drop-dead* (*drd*) are sterile, but the underlying cause of the sterility remained unknown. In this study, we examined the role of *drd* in eggshell synthesis. We show that eggs laid by *drd* mutant females are fertilized but arrest early in embryogenesis, and that the innermost layer of the eggshell, the vitelline membrane, is abnormally permeable to dye in these eggs. In addition, the major vitelline membrane proteins fail to become crosslinked by nonreducible bonds, a process that normally occurs during egg activation following ovulation, as evidenced by their solubility and detection by Western blot in laid eggs. In contrast, the Cp36 protein, which is found in the outer chorion layers of the eggshell, becomes crosslinked normally. To link the *drd* expression pattern with these phenotypes, we show that *drd* is expressed in the ovarian follicle cells beginning in mid-oogenesis, and, importantly, that all *drd* mutant eggshell phenotypes could be recapitulated by selective knockdown of *drd* expression in the follicle cells. To determine whether *drd* expression was required for the crosslinking itself, we performed *in vitro* activation and crosslinking experiments. The vitelline membranes of control egg chambers could become crosslinked either by incubation in hyperosmotic medium, which activates the egg chambers, or by exogenous peroxidase and hydrogen peroxide. In contrast, neither treatment resulted in the crosslinking of the vitelline membrane in *drd* mutant egg chambers. These results indicate that *drd* expression in the follicle cells is necessary for vitelline membrane proteins to serve as substrates for peroxidase-mediated cross-linking at the end of oogenesis.

**Funding:** This work was supported by: National Institutes of Health grant 1R15 GM080682, National Science Foundation grant IOS-1355087, and Marquette University (EMB). Marquette University Honors Program (TDS and LEK). The funders had no role in study design, data collection and analysis, decision to publish, or preparation of the manuscript.

**Competing interests:** The authors have declared that no competing interests exist.

## Introduction

Animal epithelial cells produce an extracellular matrix (ECM) that must perform many roles, including as a structural support, barrier, and source of signaling molecules [1–3]. The eggshell of the insect *Drosophila melanogaster* is a model ECM consisting of five layers of protein, lipid, and carbohydrate [4]. Among its functions, the *Drosophila* eggshell serves as physical protection and a selective permeability barrier, provides patterning signals for the oocyte and developing embryo, and binds pheromones that prevent cannibalism by conspecific larvae [5–7]. Eggshell components are primarily synthesized by the follicle cells, a layer of somatic epithelial cells that surround the 15 germline nurse cells and single oocyte; together these three cell types make up the basic unit of oogenesis, the egg chamber.

Early in oogenesis, the follicle cells form a morphologically uniform, cuboidal epithelium around the germline cells, with the apical surface of the epithelium facing the germline. At mid-oogenesis (stage 8 of the 14 stages of oogenesis), the oocyte begins to grow, and the large majority of follicle cells subsequently migrate posteriorly to cover the oocyte surface and undergo a transition from cuboidal to columnar morphology [8, 9]. A small number of follicle cells, the stretch cells, remain covering the nurse cells and transition to a squamous morphology; the stretch cells are required for the death and engulfment of the nurse cells in late oogenesis [10]. At the end of oogenesis (stage 14), the oocyte takes up the entire egg chamber, the follicle cells undergo cell death, and the follicle cell layer is shed during ovulation [11].

The innermost layer of the eggshell is the proteinaceous vitelline membrane (VM). It is composed of six related structural proteins encoded by the genes *Vm26Aa*, *Vm26Ab*, *Vm26Ac*, *Vm32E*, *Vm34Ca*, and *Vml*, as well as other less abundant proteins [4, 12–16]. While most VM components are produced by the follicle cells and secreted apically, at least three proteins, encoded by *fs(1)Nasrat* (*fs(1)N*), *fs(1)polehole* (*fs(1)ph*) and *closca* (*clos*), are secreted by the oocyte and become incorporated into the developing VM [17–19]. VM components are synthesized during mid-oogenesis (stages 8–11), followed by components of the multilayered outer sections of the eggshell, the chorion, in stages 11–12 [4, 20, 21].

Following their synthesis and secretion, the proteins of the VM become cross-linked, forming a stable and insoluble matrix. The VM proteins are cross-linked to each other by disulfide bonds during the early stages of eggshell formation [15, 22]. Immediately following ovulation and egg activation, VM proteins become cross-linked by non-reducible bonds, at least some of which are dityrosine bonds [23]. The non-reducible cross-linking of the VM occurs in a matter of minutes as the egg moves down the oviduct; soluble VM proteins are never detected in freshly laid eggs [24, 25]. While the formation of dityrosine bonds is typically catalyzed by a peroxidase [26–29], the enzyme responsible for VM crosslinking has not been identified.

The structural integrity of the *Drosophila* VM can be disrupted by mutations in several genes. Mutation of many of the genes encoding VM structural proteins causes gross VM abnormalities and collapse of the eggs [30–32], as do mutations in the cadherin *Cad99C* [33, 34], which is localized to microvilli on the apical surface of the follicle cells, and the eggshell components *yellow-g* and *yellow-g2* [12, 35]. Other mutations, in the genes encoding the minor eggshell components Nudel, Palisade (Psd), Clos, Fs(1)ph, and Fs(1)N, result in a disruption in VM protein cross-linking without altering overall VM integrity to the extent of causing eggs to collapse [17, 18, 24, 36, 37], however all of these mutations result in female sterility.

In this paper, we studied the role of the *drop-dead* (*drd*) gene in oogenesis. *drd* encodes a putative integral membrane protein of unknown function with homology to prokaryotic acyltransferases [38]. Mutation of *drd* causes a wide range of phenotypes, including female sterility, early adult death and neurodegeneration, defective food movement through the gut, and

absence of a peritrophic matrix from the midgut [39–43]. The basis for female sterility has not previously been reported. Here we demonstrate that *drd* expression in the follicle cells is required for non-reducible cross-linking of the VM.

## Materials and methods

### *Drosophila* stocks and maintenance

All fly stocks were maintained on standard cornmeal-yeast-agar food (http://flystocks.bio. indiana.edu/Fly_Work/media-recipes/molassesfood.htm) at 25˚C on a 12h:12h light-dark cycle. For RNAi experiments, a *UAS-Dcr-2* transgene was included in the genetic background of the flies in order to boost RNAi efficiency; the $drd^{GD15915}$ *UAS-Dcr-2* and *UAS-Dcr-2* $drd^{GD3367}$ lines were created previously by recombination between VDRC stocks $w^{1118}$;*P {GD3367}v37404* (FBst0461992) and $w^{1118}$; *P{GD15915}v51184* (FBst0469325) and Blooming-ton stock $w^{1118}$; *P{UAS-Dcr-2.D}2* (FBst0024650, RRID:BDSC_24650) [44, 45]. The *w\*; P {w^{+mW.hs}GAL4 = GawB}CY2* stock (FBti0007266, referred to as *CY2-GAL4*) was provided by Dr. Celeste Berg. Other stocks ($w^{1118}$; *P{UAS-GFP.nls}14* (FBst0004775, RRID:BDSC_4775), *P {w^{+mW.hs} = GawB}T155* (FBst0005076, RRID:BDSC_5076, referred to as *T155-GAL4*), *w\* ovo^{D1} v^{24} P{w^{+mW.hs} = FRT(w^{hs})}101/C(1)DX, y^1 f^1; P{ry^{+t7.2} = hsFLP}38* (FBst0001813, RRID: BDSC_1813), and *y^1 w\* v^{24} P{w^{+mW.hs} = FRT(w^{hs})}101* (FBst0001844, RRID:BDSC_1844)), were obtained from the Bloomington *Drosophila* Stock Center. Creation of the *drd-GAL4* driver transgene has been reported previously [40]. The genes and alleles referenced in this work include *drd* (FBgn0260006), $drd^{lwf}$ (FBal0193421), $drd^1$ (FBal0003113). Stocks were not outcrossed prior to this study.

### $drd^1$ sequencing

Whole-fly RNA was prepared from Canton S and $drd^1$ homozygous adults using Trizol reagent (ThermoFisher Scientific, Waltham, MA). RNA was treated with DNase (Thermo-Fisher Scientific, Waltham, MA) and cDNA was synthesized (qScript cDNA supermix, Quantabio, Beverly, MA). Primers for amplification of the exon 8/9 junction were: `CG5652 6a 5' GAT CGC CTG GTG TTT GTT TT 3'` and `CG5652 6b 5' TTC GCT GGG GAT CAC TAA AC 3'`.

### Egg-laying assay

Groups of 1–3 homozygous $drd^1/drd^1$ or $drd^{lwf}/drd^{lwf}$ females were mated with Canton S males and placed on either regular food or food supplemented with yeast paste. Flies were transferred to new vials daily until they died, and the number of eggs laid was recorded. Because groups of three flies were assayed together in early experiments, we analyzed the data twice—once assuming that all eggs were laid by a single fly (model 1) and once assuming that egg-laying was distributed evenly among all flies in a vial (model 2). The conclusions about the proportion of flies that laid eggs were the same in both analyses.

### Analysis of embryogenesis

$drd^1$ heterozygotes and homozygotes females were mated with sibling males, and eggs were collected overnight (16.5–18.5 hr) on apple juice agar plates supplemented with yeast paste [46]. Flies were removed and the eggs were allowed to develop for an additional 2–6 hr. Eggs were then covered in halocarbon 700 oil (Sigma-Aldrich, St. Louis, MO) and scored for col-lapsed vs turgid. Turgid eggs were scored for fertilized vs unfertilized, and fertilized eggs were scored for pre-gastrulation stages vs post-gastrulation using photographs of staged embryos

[46]. Unfertilized eggs were recognized by the blotchy distribution of yolk in the cytoplasm and absence of any embryonic structures. Pre-gastrulation embryos (stages 1–5) were recognized by the homogeneous distribution of yolk in the cytoplasm and absence of surface invagination or internal embryonic structure, while post-gastrulation embryos (stages 6–16) were recognized by surface invagination or internal structure.

To test the effect of overexpression of *Dcr-2* in follicle cells, $w^{1118}$; *P{UAS-Dcr-2.D}2* flies were crossed with either *CY2-GAL4* or *T155-GAL4* flies. 0–1 day old female progeny were crossed with wild-type males in four groups of 4–8 females for each GAL4 driver (20–24 total females). After two days, flies were transferred to new vials and eggs were collected for three periods of 20–24 hours, counted, and allowed to hatch for 22–31.5 hours, after which live larvae were counted and removed from the vials. Unhatched eggs were observed for four more days, but no additional hatching was seen.

## Generation of *drd* germline clones

Germline mitotic clones mutant for *drd* were generated using the FLP/FRT-dominant female sterile technique as described [47, 48]. The $drd^{lwf}$ allele was first recombined onto the same chromosome as an FRT site by crossing *w $drd^{lwf}$* x *y w v P{FRT}101*. Following the establishment of a stock carrying this recombinant chromosome, *w $drd^{lwf}$ P{FRT}101/FM7a* females were crossed with *w $ovo^{D1}$ v P{FRT}101; hs-FLP* males. The resulting *w $drd^{lwf}$ P{FRT}101/w $ovo^{D1}$ v P{FRT}101; hs-FLP/+* embryos were heat shocked for 2 hr at 37°C to induce FLPase expression and mitotic recombination, raised to adulthood, and crossed with Canton S males to assay for fertility, as indicated by the presence of live larvae. Control embryos of the same genotype were not heat-shocked. Male progeny of the germline clones were collected and their lifespan measured to confirm the presence of the $drd^{lwf}$ mutation.

## Visualization of *drd* expression pattern

*yw drd-GAL4/FM7i-GFP* females were crossed with *w; UAS-GFP.nls* males. Female progeny were crossed with sibling or wild-type males. Ovaries from females 4–7 days post-eclosion were dissected in insect Ringers, fixed in 4% paraformaldehyde in PBS (30 minutes, room temperature), washed 3x in PBS, stained with DAPI (diamidino-2-phenylindole, Biotium, Fremont, CA) in PBS (20 μg/mL, 30 minutes, room temperature), and washed 3x in PBS. Ovaries were then separated into individual egg chambers or ovarioles and mounted in Vectashield (Vector Laboratories, Newark, CA) on slides with broken coverslips used as spacers. Samples were imaged on a Nikon A1R Confocal Microscope (Nikon, Tokyo, Japan) with NIS-Elements AR software (Nikon).

## Neutral red permeability assay

Eggs were collected on apple-juice agar plates for 2–19 hr, placed into a stainless steel mesh basket, and rinsed with PBS. Eggs were dechorionated by gently shaking in a 50% bleach solution for 3 min followed by rinsing with PBS; exposure to bleach was only 1 min for *$drd^1$/$drd^1$* and *$drd^1$/FM7c*. The dechorionated eggs were counted, stained with 5 mg/mL neutral red (VWR, Radnor, PA) in PBS for 10 min, rinsed with PBS, and scored as stained or unstained. No correlation was observed between the duration of the egg collection and the staining results. See S1 File for detailed protocol.

## Western blot analysis

Egg chambers were dissected in PBS or eggs were collected on apple-juice agar plates, and samples were homogenized in 80μL of 20mM Tris-HCl (pH7.5), 0.15 M NaCl, 100 mM DTT. Samples were then heated at 100˚C for 5 min, centrifuged (14,000g, 1 min), and the resulting pellet discarded. One quarter volume 5x SDS-PAGE loading buffer (0.312 M TRIS-HCl pH 6.8, 10% SDS, 0.05% bromophenol blue) was added to each sample, and they were again heated for 5 min at 100˚C and stored at -20˚C until further use. Prior to electrophoresis, samples were treated with 5% β-mercaptoethanol and heated for 3 min at 100˚C. For any gel, to standardize the amount of protein loaded per lane, the same amount of egg chamber equivalents of each sample was used (2–4 egg chamber or eggs/lane, see figure legends for experiment-specific details). Following separation via SDS-PAGE (12%, Mini PROTEAN 3 System, Bio-Rad, Hercules, CA), proteins were transferred to PVDF membrane for 1hr using a Genie electroblot chamber (Idea Scientific, Minneapolis, MN). Membranes were then washed for 10 min in PBS and blocked overnight in PBS/0.05% Tween-20 (PBS-T)/ 5% nonfat dry milk at 4˚C. After blocking, two 5 min washes in PBS-T were conducted prior to 1 hr incubation in primary antibody (Vm26Ab, 1:10,000–25,000; Cp36 1:5000) diluted in PBS-T/1% BSA. Primary polyclonal rabbit antibodies were provided by Dr. Gail Waring and were previously characterized antibodies against Vm26Ab and Cp36 [13].

Membranes were then washed in PBS-T, once for 15 min, and four times for 5 min, followed by a 1 hr incubation in secondary antibody (ECL HRP-linked donkey anti-rabbit IgG, 1:10,000, Cytiva Life Sciences, Marlborough, MA). Again one 15 min and four 4 min washes in PBS-T were conducted and antibody signals were detected via chemiluminescence (ECL Prime Western Blotting System, Cytiva Life Sciences, Marlborough, MA).

## Immunostaining

*drd¹* heterozygous and homozygous females were collected on the day of eclosion and placed on yeast paste with sibling males for two days. Ovaries were immunostained as described [19], except that fixation was performed with 4% paraformaldehyde in PBS rather than formaldehyde in PBS/Triton X-100. Anti-Vm26Ab antibody (same as used in Western blots above) was used at 1:5000, and the secondary antibody was Alexafluor 488 goat anti-rabbit IgG (1:400) (Invitrogen, Carlsbad, CA). Samples were imaged on a Nikon A1 Confocal Microscope (Nikon, Tokyo, Japan) with NIS-Elements AR software (Nikon). VM staining intensity was determined in stage 9 and 10A egg chambers by measuring mean pixel brightness in a 1 μm x 5 μm rectangle overlaying the VM in the anterior lateral margin of the oocyte within the egg chamber (ImageJ v2.9.0), and the analyzer was blind to the genotype shown in each image. The images used for intensity measurements were all taken in one experiment with the same acquisition settings, eliminating the need for normalization of intensity measurements among different experiments. Within each genotype, there was no significant correlation between either VM width or staining intensity and developmental stage (as measured by the length/ width ratio of the oocyte within the egg chamber), allowing us to pool data across developmental stages for comparison between genotypes (S5 Fig). A small number of additional images were taken in a different experiment and were used only for measuring oocyte length and width, hence the difference in n between Figs 5C and S5.

## *In vitro* egg activation

To stimulate egg production, *drd¹/FM7c* and *drd¹/drd¹* females were placed on yeast paste and mated with sibling males 3–5 days before dissection. Egg activation *in vitro* was performed using the method of Page and Orr-Weaver [49]. Stage 14 egg chambers were dissected in

isolation buffer (55 mM NaOAc, 40 mM KOAc, 110 mM sucrose, 1.2 mM $MgCl_2$, 1 mM $CaCl_2$, 100 mM Hepes, pH 7.4 (NaOH)). Egg chambers were then incubated for 10 min in hypo-osmotic activating buffer (3.3 mM $NaH_2PO_4$, 16.6 mM $KH_2PO_4$, 10 mM NaCl, 50 mM KCl, 5% PEG 8000, 2 mM $CaCl_2$, pH 6.4 (1:5 NaOH: KOH)) and then transferred into modified Zalokar's buffer for 30 min (9 mM $MgCl_2$, 10 mM $MgSO_4$, 2.9 mM $NaH_2PO_4$ 0.22 mM NaOAc, 5 mM glucose, 27 mM glutamic acid, 33 mM glycine, 2 mM malic acid, 7 mM $CaCl_2$, pH 6.8 (1:1 NaOH: KOH)). To test for eggshell crosslinking, egg chambers were then incubated in 50% bleach for 5 min and scored as intact, leaky, or completely dissolved. In some experiments, hydrogen peroxide (0.006–0.06%) was included in both the activating and Zalokar's buffer.

In a second series of experiments, the activating buffer incubation was omitted. Egg chambers were dissected and manually dechorionated in isolation buffer and then incubated for 30 min in Zalokar's buffer containing 4.5% hydrogen peroxide and 1 mg/ml horseradish peroxidase (VWR, Radnor, PA) before bleaching as above.

## Data analysis and statistics

Data were graphed and analyzed using GraphPad Prism v9 for Windows (GraphPad Software, San Diego, CA, www.graphpad.com). A p value of <0.05 was used as the threshold for statistical significance. For the embryonic development and neutral red experiments, contingency tests were used to test for differences in binary outcomes (i.e. fertilized vs unfertilized or stained vs unstained). The Fisher exact test was used when comparing two genotypes and the Chi-square test was used when comparing more than two genotypes. For analysis of eggs laid per female (S2 Fig) and immunostaining intensity (Fig 5), each dataset was tested for normal distribution using the D'Agostino & Pearson normality test. For the egg-laying data, two datasets were not normally distributed ($drd^{lwf}$ model 2 and yeast-fed $drd^1$), and so comparisons among genotypes and conditions were done using the non-parametric Kruskal-Wallis and Dunn's multiple comparison tests. For the immunostaining data, the VM width datasets were both normally distributed and so were compared using a parametric t-test. The heterozygous VM intensity dataset was not normally distributed and so intensities were compared using a non-parametric Mann-Whitney test.

## Results

### Identification of the $drd^1$ mutation

The experiments in this study utilize flies carrying the two most severe alleles of *drd*: $drd^{lwf}$ and $drd^1$. The mutation in the latter of these alleles has not been molecularly characterized, although we previously reported that there were no alterations in the protein coding sequence [38]. Further sequencing of the final 5 introns and the ends of the first three large introns revealed six differences between $drd^1$ and wild-type. One of these, a T to A transversion in the final intron, is predicted to create a strong ectopic splice acceptor site and result in the inclusion of an additional 10 nucleotides in the spliced transcript (S1 Fig) (www.fruitfly.org/seq_tools/splice.html) [50]. This aberrant splicing of the *drd* transcript in $drd^1$ mutants was confirmed by PCR on whole-fly cDNA (S1 Fig). Virtual translation of the mutant transcript predicted that the final 76 amino acids of the Drd protein are replaced with a novel sequence of 45 amino acids in the $drd^1$ mutant.

## Sterility of *drd* mutant females

We have previously reported that females homozygous for severe *drd* alleles are sterile and rarely lay eggs, and their ovaries contain very few vitellogenic egg chambers [38]. We hypothesized that these phenotypes could be an effect of the starvation observed in *drd* mutant flies, as opposed to a direct phenotype of *drd* mutation. Consistent with this hypothesis, feeding females a high protein diet (yeast paste) stimulated egg-laying in a subset of flies. Yeast paste increased the percentage of *drd^{lwf}* females, but not *drd^1* females, that laid eggs, and it increased by more than 30-fold the median number of eggs laid during their lifetime by those *drd^1* and *drd^{lwf}* homozygous females that laid eggs (S2 Fig).

Despite the improvement in egg-laying, *drd* homozygous females fed with yeast paste remained sterile. As shown in Table 1, a significantly higher number of eggs laid by *drd^1* homozygotes were collapsed compared with eggs laid by heterozygote controls (p<0.0001, Fisher's exact test). Of the turgid eggs, the large majority were fertilized, and the rate of fertilization was not affected by the mother's genotype (p = 0.30, Fisher's exact test). We did not quantify any other aspects of egg morphology, such as overall dimensions or size of the dorsal appendages. Virtually no fertilized eggs laid by homozygotes underwent gastrulation, in contrast to embryos laid by heterozygotes (p<0.0001, Fisher's exact test). Thus, the sterility of *drd* mutant females appears to result from early embryonic arrest.

## Expression of *drd* in the egg chamber

To determine whether *drd* expression was required in the germline or soma for female fertility, we used the FLP-FRT dominant female sterile technique to create females with a *drd* mutant germline [48]. This technique uses the X-linked dominant female sterile mutation *ovo^{D1}*, which prevents formation of vitellogenic egg chambers. We created flies that were heterozygous for *drd^{lwf}* and *ovo^{D1}*, with the two mutations on different X chromosomes. Both X-chromosomes in these flies also contained a FRT recombination site, and the flies had a heat-shock inducible FLP recombinase. In the absence of heat-shock, all cells in these females will be heterozygous for *drd^{lwf}* and *ovo^{D1}*, and the females should be sterile. Heat-shock induced FLP expression during embryogenesis would cause mitotic recombination in some cells, resulting in cells that are homozygous for either *drd^{lwf}* or *ovo^{D1}*. Female fertility would only be observed with germline cells that had lost *ovo^{D1}* and were homozygous for *drd^{lwf}*, and only if *drd* was not required in the germline. We observed that 15 of 17 heat-shocked females were fertile when crossed with wild-type males, while non-heat shocked controls were all sterile (n = 28), indicating that *drd* expression is not required in the female germline for fertility. The male

**Table 1. Sterility of *drd* mutant females.**

| Maternal genotype (# of eggs) | *drd^1*/FM7c (164) | *drd^1*/ *drd^1* (293) |
|---|---|---|
| Collapsed (% of total) | 0.6% | 17.7%**** |
| Unfertilized (% of turgid eggs) | 7.5% | 11.2% |
| Pre-gastrulation (% of fertilized eggs) | 6.8% | 99.0%**** |
| Post-gastrulation (% of fertilized eggs) | 93.2% | 1.0%**** |

Table 1: Embryonic arrest in eggs laid by *drd^1* females. The second row indicates the percentage of all eggs that were collapsed or flaccid. The following row indicates the percentage of turgid eggs that were unfertilized (4 eggs from heterozygotes and 9 from homozygotes could not be scored and were excluded). The final two rows indicate the percentage of fertilized eggs that were scored as pre-gastrulation and post-gastrulation, respectively. Fisher's exact test:

****: different from sibling controls, p<0.0001.

progeny of these clones were short-lived (median lifespan of 4 days, n = 55), as would be expected for male progeny of a *drd^lwf* homozygous germline.

Consistent with the germline clone analysis, driving a GFP reporter with a *drd-GAL4* transgene resulted in labeling of ovarian follicle cells but not the germline (Figs 1 and S3). No reporter expression was observed in egg chambers prior to stage 10B (S3A and S3B Fig). At stage 10B, expression was observed in most follicle cells, although some appeared to be unlabeled (Fig 1A), and at later stages all follicle cells appeared to be labeled (Figs 1B and S3C). Labeling was observed in main body follicle cells covering the oocyte, in centripetal cells between the oocyte and nurse cells, and in stretch cells covering the nurse cells (Figs 1A and S3C and S3D). Control egg chambers lacking the *drd-GAL4* driver did not show GFP expression (S3E and S3F Fig).

## Abnormal eggshell development in *drd* mutant females

Because a significant number of eggs laid by *drd* mutants were collapsed and because *drd* expression was observed in the follicle cells and not the germline, we next examined the integrity of the eggshell, a structure synthesized by the somatic follicle cells. The integrity of the inner layer of the eggshell, the VM, was assayed by staining dechorionated eggs with neutral red, a dye that is normally excluded by the VM [24]. Virtually no eggs laid by *drd^1* and *drd^lwf* heterozygotes lysed during dechorionation (1–3%) or were stained with neutral red (3–4%) (Fig 2). In contrast, eggs laid by homozygotes of either *drd* allele showed a high susceptibility to lysis during dechorionation (31–42%), and the large majority of surviving eggs were permeable to neutral red (71–90%). A small fraction of eggs (4–11%) were not dechorionated upon treatment with bleach and could not be assessed for dye exclusion, but the abundance of such eggs was not affected by the genotype of the mother (p = 0.19, Chi-square test).

As a direct test of the incorporation of eggshell proteins into an insoluble cross-linked matrix, we performed Western blots on lysates of staged egg chambers and laid eggs. An antibody against the chorion protein Cp36 detected the expected pattern of staining in samples from *drd^1/FM7c* heterozygous females (Fig 3) [13]. Soluble Cp36 protein was not detected at stage 10 of oogenesis, which is before chorion proteins such as Cp36 begin to be expressed, but Cp36 was detected in stage 14 lysates, when the protein is present in the chorion but has not yet been fully cross-linked. In lysates from eggs collected either 3 or 6 hr after deposition, Cp36 was not detectable, consistent with complete cross-linking of the protein into the insoluble chorion. The same pattern of staining was observed in lysates from eggs and egg chambers from *drd^1* homozygotes (Fig 3), indicating that crosslinking of the chorion, or at least of the specific protein Cp36, is not affected by mutation of *drd* (one biological replicate performed).

In contrast to the chorion, crosslinking of VM proteins was clearly abnormal in eggs laid by *drd* mutant homozygotes. Fig 4A shows the pattern of staining observed in egg chambers and eggs from heterozygous controls, using an antibody raised against the VM protein Vm26Ab [13]. This antibody has been reported to cross-react with multiple VM proteins due to their high degree of sequence similarity [22, 51], and we typically observed multiple bands on Western blots. As expected, staining was observed in lysates from stage 10 egg chambers of heterozygous controls, when the VM proteins are not yet crosslinked, and from stage 14, when the VM proteins are crosslinked by disulfide bonds but are soluble in the presence of reducing agents. In laid eggs from heterozygotes, no soluble VM proteins were observed, consistent with the formation of nonreducible dityrosine bonds among VM proteins during ovulation and egg activation. In lysates of laid eggs from *drd^1* homozygotes, VM proteins were detected on the Western blot, indicating that these proteins are not fully crosslinked in the absence of *drd* expression. We observed solubility of VM proteins in laid eggs of *drd^1* homozygotes in

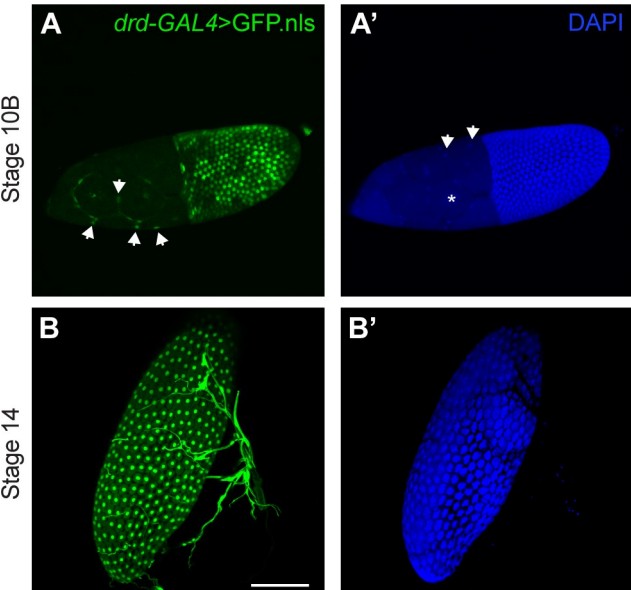

**Fig 1. Expression of *drd* in ovarian follicle cells.** (A, B) GFP expression driven by the drd-GAL4 driver. (A′, B′) DAPI staining of cell nuclei. (A, A′) A stage 10B egg chamber, showing GFP labeling of most follicle cell nuclei. The posterior end of the egg chamber, containing the majority of follicle cells and the oocyte, is to the right, and the anterior end, containing the nurse cells and stretch follicle cells, is to the left. The asterisk in A′ indicates one of the large nurse cell nuclei, which are not labeled with GFP. Arrows in A indicate labeled stretch follicle cell nuclei, and arrow in A′ indicate unlabeled stretch follicle cell nuclei. (B, B′) A stage 14 egg chamber, showing GFP labeling of all follicle cell nuclei. The structures to the right of and overlapping with the egg chamber are respiratory tracheae, which also express *drd*. Images are maximum intensity projections of Z-stacks. Scale bar is 100 μm.

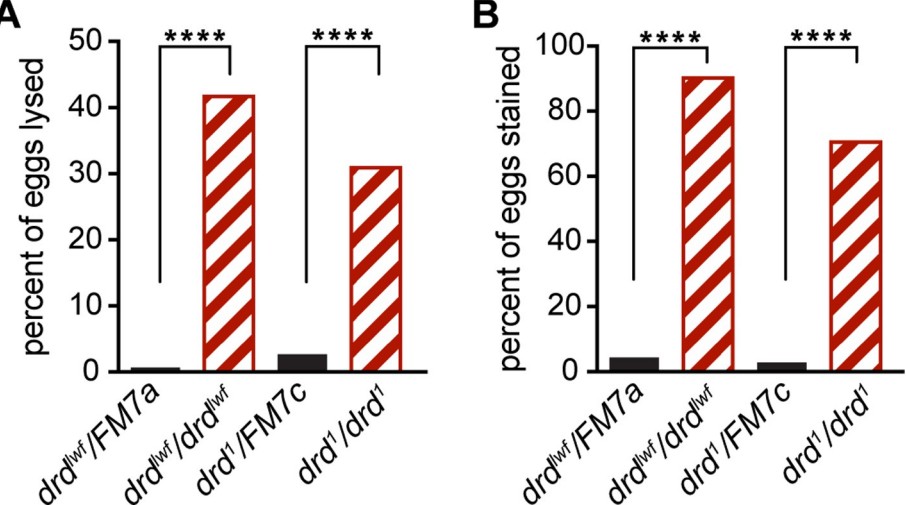

**Fig 2. Tests of eggshell assembly in eggs laid by *drd* homozygous and heterozygous females.** (A) fraction of eggs that lysed during bleach treatment. (B) fraction of eggs that were stained with neutral red after successful dechorionation. ****: significant difference between eggs laid by heterozygotes vs homozygotes, p<0.0001, Fisher's exact test. Black solid bars: eggs laid by heterozygous control females; red striped bars: eggs laid by homozygous mutant females. The starting number of eggs was 139 from *drd*$^{lwf}$/*FM7a* females, 115 from *drd*$^{lwf}$/*drd*$^{lwf}$, 225 from *drd*$^{1}$/*FM7c*, 249 from *drd*$^{1}$/*drd*$^{1}$.

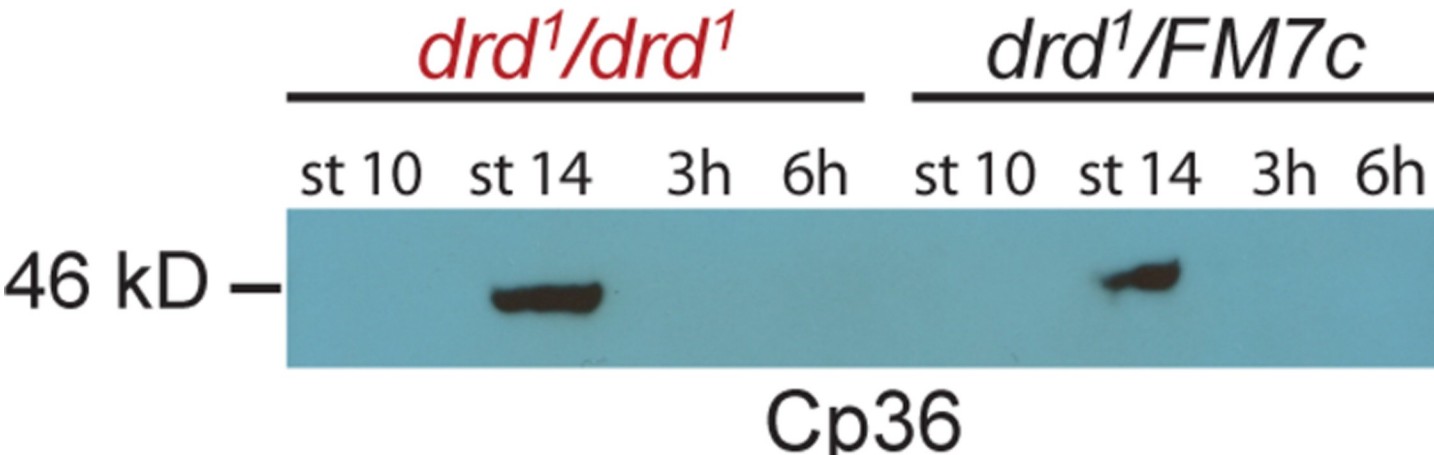

**Fig 3. Western blot of the chorion protein Cp36 in *drd* mutant and control egg chambers and eggs.** Stage 10 and 14 egg chambers were dissected from $drd^1/drd^1$ mutant and $drd^1/FM7c$ heterozygous females, and eggs laid by these females were collected 0–3 and 0–6 hr after oviposition (lanes 3 and 7 and lanes 4 and 8). 4 eggs or egg chambers per lane, 1:5000 Cp36 primary antibody dilution.

seven Western blots representing five biological replicates. Only two of these experiments included multiple intervals between oviposition and egg collection, and the decrease in signal seen in Fig 4A with increasing time after oviposition (compare lanes 3 and 4) was not observed in the second such experiment. S4A Fig shows an overexposed blot indicating the presence of protein in the lane loaded with laid eggs from heterozygotes (lane 4).

To assay for the formation of disulfide bonds among VM proteins during oogenesis, lysates were prepared from stage 10 and 14 egg chambers in the absence of reducing agents. Mutation of *drd* did not alter the pattern of immunostaining observed under these conditions (Fig 4B). Soluble protein was detected at stage 10 but not at stage 14, indicating that VM proteins are

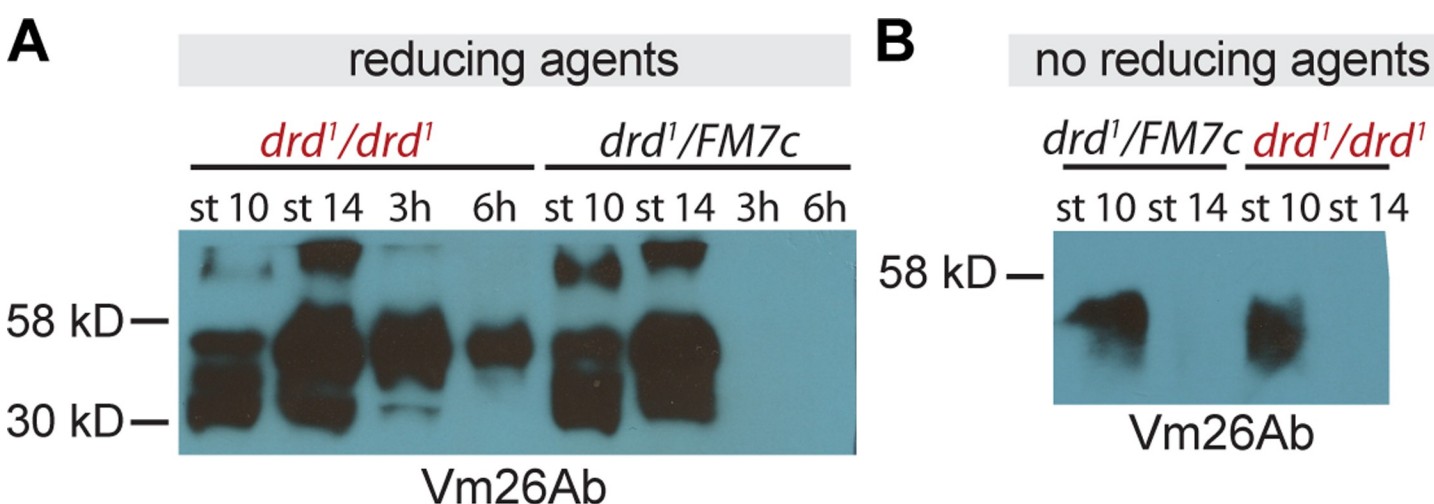

**Fig 4. Western blots against vitelline membrane proteins in *drd* mutant egg chambers and eggs.** (A) Western blot with an antibody raised against Vm26Ab of samples treated with reducing agents. Samples include stage 10 and 14 egg chambers dissected from $drd^1/drd^1$ mutant and $drd^1/FM7c$ heterozygous females, as well as eggs laid by these females that were collected 0–3 and 0–6 hr after oviposition. 4 eggs or egg chambers per lane, 1:10,000 primary antibody dilution. (B) Western blot from egg chambers solubilized in the absence of reducing agent and probed with an antibody against Vm26Ab. Samples include stage 10 and 14 egg chambers dissected from $drd^1/drd^1$ mutant and $drd^1/FM7c$ heterozygous females. 2 egg chambers per lane, 1:10,000 Vm26Ab primary antibody dilution.

cross-linked by reducible bonds during oogenesis in both *drd* homozygotes and heterozygotes (one biological replicate performed).

We performed immunostaining of stage 9 and 10A egg chambers against Vm26Ab to determine whether the appearance of the developing VM is altered in *drd* mutants (Fig 5A and 5B). We observed no significant effect of genotype on either the staining intensity (p = 0.14, Mann-Whitney test) or width (p = 0.54, t-test) of the VM (S5 Fig, n = 21 heterozygote and 22 homozygote). There were also no obvious morphological differences in the egg chambers between the two genotypes. However, *drd¹* homozygous egg chambers were slightly but significantly smaller than heterozygous controls. We used the size ratio of anterior-posterior length/lateral width of the oocyte within each egg chamber as a measure of progression through oogenesis, as this ratio increases during stages 9 and 10A (Fig 5C). Anterior-posterior length was significantly linearly correlated with the size ratio for each genotype (p<0.0001, F test). The slope of the relationship did not differ between the two genotypes (p = 0.88, F test) but the intercept did differ (p = 0.02, F test), corresponding to a decrease in size of homozygous oocytes of approximately 15%.

## Follicle cell knockdown of *drd* recapitulates mutant phenotypes

Our finding that *drd* expression is required in the soma for female fertility, coupled with the known role of the somatic follicle cells in the synthesis of the eggshell, suggested that all of the *drd* mutant phenotypes related to fertility and eggshell assembly could be associated with the expression of *drd* in the follicle cells. To test this, we knocked down *drd* expression in the follicle cells, using two pan-follicle cell GAL4 drivers, *CY2-GAL4* and *T155-GAL4*, and two inducible *drd* RNAi transgenes, *drd^{GD3367}* and *drd^{GD15915}*, that we have previously shown to be effective at knocking down *drd* expression [44]. Females heterozygous for a GAL4 driver and an RNAi transgene (referred to as *drd* knockdown females) and sibling control females that lacked the RNAi transgene were mated with sibling males, and their eggs were scored for fertilization, embryonic development, eggshell integrity, and crosslinking. As shown in Table 2, a

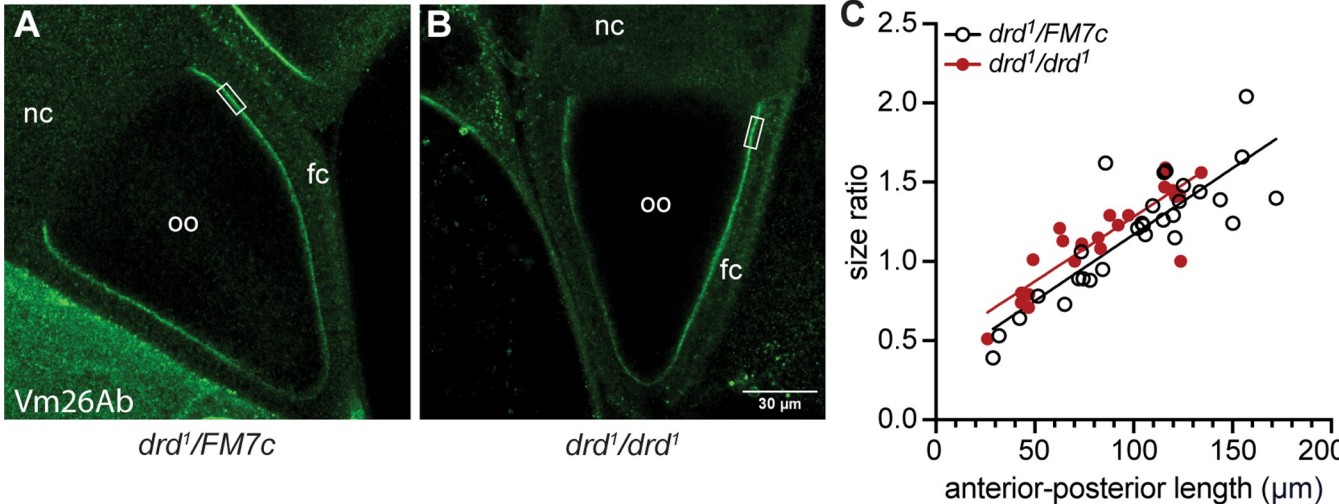

**Fig 5. Immunostaining of the VM.** Stage 10A egg chambers from a *drd¹* heterozygote (A) and homozygote (B) were stained with the anti-Vm26Ab antibody. The rectangle indicates the region in which staining intensity and eggshell width were measured. Oo: oocyte; nc: nurse cells; fc: follicle cells. (C) Plot of the size ratio of anterior-posterior length/lateral width for each oocyte as a function of the anterior-posterior length. The lines are the result of simple linear regression of each dataset, showing the slight difference in Y-intercept between *drd¹* heterozygotes and homozygotes (F test, p<0.0001 for significant linear correlation of each dataset, p = 0.88 for comparison of slopes, p = 0.02 for comparison of Y-intercepts). n = 29 heterozygotes and 25 homozygotes.

**Table 2. Sterility of *drd* knockdown females.**

| Maternal genotype (# of eggs) | *w; CyO/+; T155-GAL4/+* (153) | *w; CyO/+; T155-GAL4/+* (221) | *w; CY2-GAL4/CyO* (238) | *w; CY2-GAL4/CyO* (163) |
|---|---|---|---|---|
| Collapsed (% of total) | 0% | 0.5% | 0.4% | 0.6% |
| Unfertilized (% of turgid eggs) | 3.4% | 10.3% | 2.2% | 4.5% |
| Pre-gastrulation (% of fertilized eggs) | 0.7% | 1.0% | 1.3% | 2.7% |
| Maternal genotype (# of eggs) | *w; UAS-Dcr-2 drd$^{GD3367}$/+; T155-GAL4/+* (188) | *w; drd$^{GD15915}$ UAS-Dcr-2/+; T155-GAL4/+* (155) | *w; UAS-Dcr-2 drd$^{GD3367}$/ CY2-GAL4* (169) | *w; drd$^{GD15915}$ UAS-Dcr-2/ CY2-GAL4* (206) |
| Collapsed (% of total) | 32.4%**** | 43.9%**** | 21.9%**** | 24.3%**** |
| Unfertilized (% of turgid eggs) | 6.1% | 10.0% | 7.8%* | 4.8% |
| Pre-gastrulation (% of fertilized eggs) | 100%**** | 98.6%**** | 100%**** | 96.4%**** |

Table 2: Embryonic arrest upon knockdown of *drd* expression in the follicle cells. Sibling control females (top four rows) and *drd* knockdown females (bottom four rows) were mated with sibling males, and eggs were collected and scored as in Table 1. The number of uncollapsed but unscorable eggs that were excluded from further analysis varied from 4–13 among each of the eight genotypes. Fisher's exact test:

****: different from sibling controls, p<0.0001

*: p = 0.024.

greater percentage of eggs from *drd* knockdown females were collapsed compared with sibling controls, and virtually all fertilized eggs from *drd* knockdown females were arrested pre-gastrulation, as was seen in mutant females (Fisher's exact test, p<0.0001 for each genotype). The proportion of eggs laid by *drd* knockdown females that was fertilized was not significantly different than that of sibling controls for three of the four genotypes tested, with a small but significant effect on fertilization for the combination of the *CY2-GAL4* driver and the *drd$^{GD3367}$* RNAi transgene. Despite the near-universal arrest of eggs from *drd* knockdown females early in oogenesis in this assay, we did occasionally observe larvae in vials of knockdown females carrying *T155-GAL4* and either of the RNAi-transgenes, indicating that these females were not fully sterile. The *drd* knockdown females carrying the *CY2-GAL4* driver, like *drd* mutant females, appeared to be completely sterile.

To control for any disruption of oogenesis by overexpression of *Dcr-2* in the follicle cells, we measured hatching rates of eggs laid by females in which *UAS-Dcr-2* expression was driven by either *CY2-GAL4* or *T155-GAL4*. In both cases, 98.4% of laid eggs successfully hatched (363/369 for *CY2-GAL4* and 249/253 for *T155-GAL4*), indicating that the sterility shown in Table 2 is not due to overexpression of *Dcr-2*.

Tests of eggshell integrity by neutral red exclusion demonstrated that eggs laid by *drd* follicle cell knockdown females showed a similar defect to those laid by *drd* mutants. As shown in Fig 6, the percentage of eggs that lysed upon dechorionation was significantly different between knockdown and sibling control females for three of the four genotypes (Fig 6A). In contrast to the data from *drd* mutants, we observed a small but significant increase (6–12%) in the fraction of eggs from *drd* knockdown females that were successfully dechorionated relative to sibling controls when the *drd$^{GD3367}$* RNAi transgene was driven by either of the two GAL4 lines (Fig 6B). Importantly, we observed a significant and consistent increase in the proportion of eggs from *drd* knockdown females that were permeable to neutral red compared to those laid by sibling controls (Fig 6C).

Crosslinking of VM proteins in laid eggs and stage 14 egg chambers from knockdown and control females was assayed by Western blot with the antibody against Vm26Ab as described

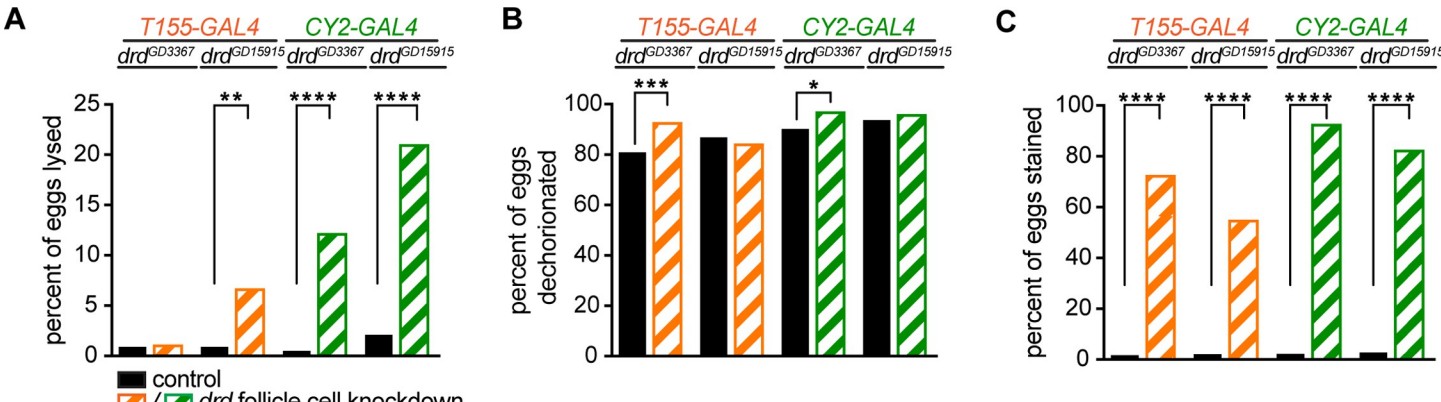

**Fig 6. Tests of eggshell assembly following *drd* knockdown.** Data from eggs laid by *drd* knockdown (striped orange bars for *T155-GAL4* and striped green bars for *CY2-GAL4*) and sibling control (solid black bars) females. (A) Fraction of eggs that lysed during bleach treatment. (B) Fraction of eggs that were successfully dechorionated. (C) Fraction of eggs that were stained with neutral red after dechorionation. Significant difference between eggs laid by *drd* knockdown females and sibling controls by Fisher's exact test are indicated: **** p<0.0001, *** p = 0.0006, ** p = 0.0016, * p = 0.015. The starting number of eggs per genotype, reading left to right on the graphs, was 224, 200, 225, 213, 206, 199, 195, 225.

above. A representative blot is shown in Fig 7. The *drd* knockdown females carrying the *CY2-GAL4* driver showed a phenotype identical to that of *drd* mutants: we consistently observed soluble VM proteins in both stage 14 egg chambers and laid eggs, while soluble VM proteins were only observed in the stage 14 chambers of sibling controls (three of three biological replicates with *drd*[GD3367] and two of two with *drd*[GD15915]; three technical replicates were run for one set of samples for each RNAi transgene). In contrast, soluble VM proteins were detected in only some samples of eggs laid by *drd* knockdown females carrying the *T155-GAL4* driver (two of four biological replicates with *drd*[GD3367] and one of three with *drd*[GD15915]; two technical replicates were run for one set of *drd*[GD3367] samples). S4B and S4C Fig shows overexposed blots indicating the presence of protein in the lane loaded with laid eggs from controls (B, lane 5, C, lanes 2 and 6).

## Cross-linking of VM in isolated egg chambers

To determine whether the *drd* cross-linking defect persists in isolated egg chambers, we dissected stage 14 egg chambers and activated them *in vitro* by exposure to hypo-osmotic medium as previously reported [49]. VM cross-linking was assayed by then incubating the egg chambers in 50% bleach (Fig 8A). All egg chambers dissected from *drd*[1]/FM7c control females were successfully cross-linked following hypo-osmotic treatment (three independent trials). In contrast, all egg chambers dissected from *drd*[1] homozygotes completely dissolved in bleach (three independent trials). Additional egg chambers were dissected from homozygotes and treated with 0.006–0.06% hydrogen peroxide during treatment with hypo-osmotic medium and for 30 min afterwards, and all these egg chambers also dissolved completely in bleach (Fig 8A, right bar, three independent trials). It is noteworthy that the *drd* mutant phenotype is more severe in egg chambers activated *in vitro* than *in vivo*, as most eggs laid by *drd* homozygous females remain intact upon treatment with bleach (see Fig 2 above).

We then attempted to directly cross-link the VM of stage 14 egg chambers by treatment with hydrogen peroxide and peroxidase. Stage 14 egg chambers were dissected, manually dechorionated, and treated with peroxide/peroxidase without exposure to hypo-osmotic medium (Fig 8B). Egg chambers dissected from *drd*[1]/FM7c control females and treated with 1 mg/ml peroxidase and 4.5% hydrogen peroxide all survived a subsequent challenge with bleach (two

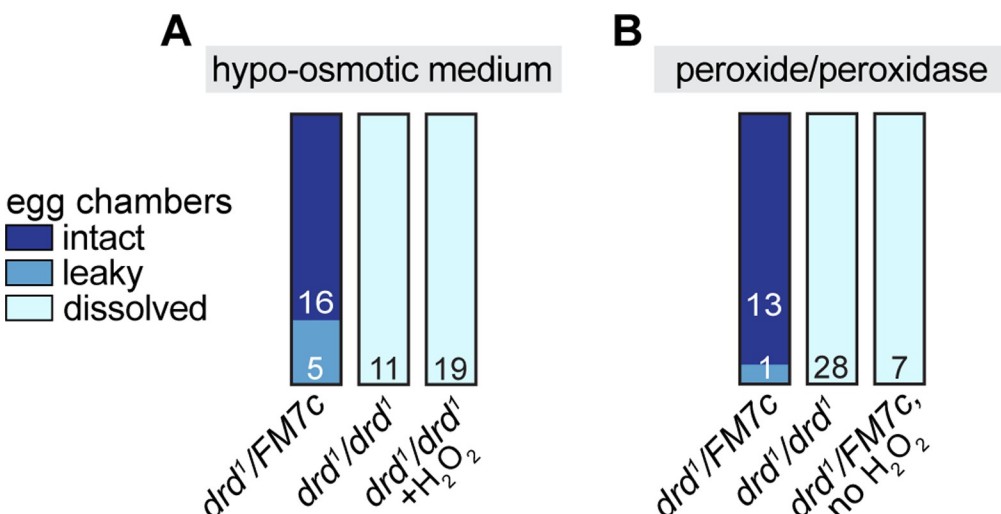

**Fig 7. Western blots against vitelline membrane proteins in *drd* follicle cell knockdown egg chambers and eggs.** Blots were probed with an antibody raised against Vm26Ab. Samples include stage 14 egg chambers dissected from, and eggs laid by, females in which the inducible *drd* RNAi transgene, *drd^GD3367^*, was driven by either the *CY2-GAL4* or *T155-GAL4* driver, or sibling controls lacking the RNAi transgene. Lanes are marked "st 14" for egg chambers and "e" for eggs. 2 eggs or egg chambers per lane, 1:25,000 Vm26Ab primary antibody dilution. Eggs were collected between 0–4.5 hr after oviposition.

independent trials). This cross-linking was not due to inadvertent activation of the egg chambers during dechorionation, as omitting the hydrogen peroxide from the incubation resulted in all egg chambers dissolving in bleach (one trial). Egg chambers from *drd^1^* homozygotes treated with peroxide/peroxidase as above all dissolved in bleach (five independent trials).

**Fig 8. Results of *in vitro* egg activation.** The figure indicates the number of egg chambers dissected from *drd^1^/drd^1^* mutant females and *drd^1^/FM7c* heterozygous sibling controls that remained intact, became leaky, or dissolved in 50% bleach after treatment with hypo-osmotic medium (A) or hydrogen peroxide and horseradish peroxidase (B).

## Discussion

We have demonstrated that expression of *drd* in the ovarian follicle cells is both necessary and sufficient for female fertility and the progression of embryonic development beyond gastrulation. We first demonstrate that *drd* is expressed in ovarian follicle cells, but not the germline. A reporter that is expected to mimic the pattern of *drd* expression was expressed in the ovarian follicle cells and not the germline. Consistent with this, the result of our dominant female sterile experiment indicates that *drd* expression in the female germline is not required for fertility, while knockdown of *drd* expression specifically in the follicle cells recapitulates the mutant phenotype. These results are in keeping with those of Kim et al, who reported expression of *drd* in the follicle cells [52]. Furthermore, we have shown that *drd* expression in the follicle cells is required for proper development of the VM layer of the eggshell. In the absence of such expression, many eggs collapse, and the remaining eggs have a fragile VM that fails to act as a permeability barrier. Several major VM proteins—specifically those recognized by the antibody used in these studies—remain soluble in the presence of reducing agents, indicating that they have not been incorporated into the insoluble network of cross-linked proteins seen in the wild-type VM. The variable solubility phenotype observed with knockdown of *drd* with the *T155-GAL4* driver is consistent with the fertility of some of these *drd* knockdown females and suggests that *T155-GAL4* is not as effective as the *CY2-GAL4* driver in knocking down *drd* expression. The difference in strength of these two driver lines has been reported previously [53].

The solubility of VM proteins in eggs laid by *drd* mutants indicates a defect in the peroxidase-mediated cross-linking that normally occurs upon egg activation while the egg is transiting down the oviduct. We have no evidence that egg activation itself is defective, as the eggs were able to complete the early stages of embryogenesis. Our data in this study don't address the connection between defective VM cross-linking and embryonic arrest. Although it is possible that embryonic arrest is a *drd* phenotype independent of the defect in VM cross-linking, it has previously been reported that defective VM cross-linking is sufficient to cause early embryonic arrest. Females mutant for *psd*, which encodes a minor VM protein, also lay eggs with a VM cross-linking defect and that arrest pre-gastrulation and show a chromatin margination phenotype similar to that induced by anoxia [36]. Similarly, females with class I alleles of the eggshell component *nudel* lay eggs that exhibit VM cross-linking defects and, though extremely fragile, are fertilized and arrest early in embryogenesis [24, 54]. Thus, it is likely that defective VM cross-linking is also the direct cause of early developmental arrest in *drd* mutants. While our study did not find a role for *drd* in embryonic development, based on our experiments, such a role cannot be ruled out.

A published microarray study of ovarian gene expression has reported that *drd* expression in the egg chamber begins at stage 8 of oogenesis, peaks at stages 10A and 10B, and then declines [20]. The timing of *drd* expression therefore parallels the synthesis and secretion of VM proteins by the follicle cells. The pattern of reporter expression shown in Figs 1 and S3 is consistent with the microarray data. We observed GFP fluorescence starting at stage 10B and persisting through the rest of oogenesis; one would expect both a delay between the onset of GFP expression and significant accumulation in the follicle cell nuclei and persistence of the protein after gene expression is downregulated. The Drd protein is unlikely to be a component of the VM, as it is predicted to be an integral membrane protein and is reported to be localized to an intracellular compartment [38, 52]. Drd is also unlikely to be directly involved in the cross-linking process, which occurs at the end of oogenesis when *drd* expression is very low. Rather, our data suggest that the failure of VM proteins to become cross-linked in *drd* mutants could be due to an absence of some modification of the VM proteins in the follicle cells prior

to secretion. The results of our final experiment are consistent with this hypothesis. Incubation of stage 14 egg chambers with peroxide and peroxidase resulted in cross-linking of the VM in egg chambers from wild-type but not *drd* mutant females. Thus, VM proteins synthesized and secreted from *drd* mutant follicle cells appear to be poor substrates for peroxidase-mediated cross-linking for reasons still to be determined.

One interesting finding from our final experiment is that egg chambers from *drd* mutant females dissolve immediately in bleach after activation *in vitro* with hypo-osmotic medium. In contrast, eggs laid by *drd* mutant females mainly survive bleaching, even though their VMs are permeable to neutral red. The contrast between these two results indicates that hypo-osmotic treatment *in vitro* does not fully recapitulate the activation process *in vivo* even though the two processes appear to give identical results in wild-type flies [49]. Additional study of oogenesis in *drd* mutant females could prove to be useful for identifying additional factors required for eggshell maturation during ovulation.

In addition to characterizing the effect of *drd* mutations on oogenesis, we have identified the molecular defect in the severe *drd*[1] allele as a point mutation in the final intron that disrupts the normal splicing of exons 8 and 9. The aberrant splicing replaces the final 76 residues of the 827 amino acid Drd protein with a novel sequence. *drd*[1] has previously been shown to cause the same short adult lifespan, female sterility, and absence of a peritrophic matrix as *drd*[lwf], a nonsense mutation that eliminates all but the first 180 amino acids [38, 40]. Thus, our finding highlights the importance of the C-terminal of Drd in protein function, stability, or localization. In contrast, the *drd*[W3] and *In(1)drd*[x1] alleles, which eliminate the first exon and at least the first 125 amino acids, are phenotypically less severe [38].

The biochemical function of the Drd protein remains unknown. However, this study highlights two themes that are emerging from our studies of this gene. First, *drd* expression appears to be required in a number of different epithelial tissues, including the ovarian follicle cells for oogenesis, the anterior midgut for digestive function, and the respiratory tracheae for brain integrity. Second, *drd* expression is required for the formation of extracellular barrier structures, as *drd* mutants show defects both in the eggshell and in the peritrophic matrix of the midgut. Given the amount of information known about eggshell formation, the ovary is an excellent system for further studies of *drd* function.

## Supporting information

**S1 File. Protocol for neutral red permeability assay.**
(PDF)

**S2 File. Raw numerical data.** The six tabs in the spreadsheet contain underlying data for S2 and S5 Figs and Tables 1 and 2 and Figs 2, 5 and 6, respectively.
(XLSX)

**S1 Fig. Mutation in the *drd*[1] allele.** (A) Genomic DNA sequences. Upper sequence shows the wild-type sequence of the final five bases of exon 8 (bold), all of intron 8, and the first five bases of exon 9 (bold). The lower sequence shows the same region in *drd*[1], with the single base change (green, underlined) and the new start of exon 9 (bold). (B) cDNA sequences. Upper sequence shows the wild-type sequence of cDNA from the region of the exon 8/9 junction. The splicing site is indicated in red. The lower sequence shows the same region in *drd*[1], with the new splice junction shown in red and the added 10 bases in bold.
(PDF)

**S2 Fig. Egg-laying by *drd* mutant females.** (A) Fraction of females in each condition that laid any eggs during their lifetime. Because some non yeast-fed females were assayed in groups of

2–3, the data were analyzed with two different assumptions regarding the distribution of egg-laying (see methods). Brackets indicate the effect of yeast feeding (two-sided Fisher's exact test). (B) Number of eggs laid per female, omitting data from those females that laid no eggs. Brackets indicate the effect of yeast feeding (Kruskal-Wallis test with Dunn's post-hoc multiple comparisons test). Blue: $drd^{lwf}$ females; Red: $drd^1$ females. n = 67 $drd^{lwf}$ females, 21 yeast-fed $drd^{lwf}$, 48 $drd^1$, 24 yeast-fed $drd^1$.
(PDF)

**S3 Fig. Expression of *drd* in ovarian follicle cells.** (A-F) GFP expression driven by the *drd-GAL4* driver. (A′-F′) DAPI staining of cell nuclei, with nurse cell nuclei in the anterior of each egg chamber indicated with asterisks. (A, A′) A stage 9 egg chamber, showing no GFP labeling. (B, B′) A stage 10A egg chamber, showing no GFP labeling. (C, C′) A stage 11 egg chamber, showing labeled follicle cells. (D, D′) A stage 10B egg chamber showing labeled follicle cells. Labeled stretch and centripetal follicle cells in C and D are indicated with arrowheads and arrows, respectively. (E, E′, F, F′) Control stage 10B (E, E′) and stage 12 (F, F′) egg chambers lacking the *drd-GAL4* transgene. A-D are single confocal images, and E-F are maximum intensity projections of Z-stacks. Scale bar is 100 μm.
(TIF)

**S4 Fig. Overexposed Western blots showing signals in "blank" lanes.** (A) Stage 10 and 14 egg chambers dissected from $drd^1/FM7c$ heterozygous and $drd^1/drd^1$ mutant females, as well as eggs (e) laid by these females that were collected 0–5 hr after oviposition. 2 eggs or egg chambers per lane (except mutant st 10, 1 egg chamber), 1:25,000 primary antibody dilution. (B) Laid eggs collected 0–4.5 hr after oviposition. Lane 1: *w; CyO/+; T155-GAL4/+*, (sibling controls of lane 2); Lane 2: *w; UAS-Dcr-2 $drd^{GD3367}$/+; T155-GAL4/+*; Lane 3: *w; CyO/+; T155-GAL4/+* (sibling controls of lane 4); Lane 4: *w; $drd^{GD15915}$ UAS-Dcr-2/+; T155-GAL4/+*; Lane 5: *w; CY2-GAL4/CyO* (sibling controls of lane 6); Lane 6: *w; UAS-Dcr-2 $drd^{GD3367}$/ CY2-GAL4*; Lane 7: *w; CY2-GAL4/CyO* (sibling controls of lane 8); Lane 8: *w; $drd^{GD15915}$ UAS-Dcr-2/CY2-GAL4*. 5 eggs per lane, 1:10,000 primary antibody dilution. (C) Stage 14 egg chambers and laid eggs collected 0–3.5 h after oviposition. Lanes 1–2: *w; CY2-GAL4/CyO* (sibling control of lanes 3–4); Lanes 3–4: *w; $drd^{GD15915}$ UAS-Dcr-2/CY2-GAL4*; Lanes 5–6: *w; CY2-GAL4/CyO* (sibling control of lanes 7–8); Lanes 7–8: *w; UAS-Dcr-2 $drd^{GD3367}$/CY2-GAL4*. Left gel, 2 eggs or egg chambers per lane; right gel, 1 egg or egg chamber per lane, 1:25,000 primary antibody dilution. Asterisks indicate signals in lanes containing laid eggs from control females.
(PDF)

**S5 Fig. VM staining intensity and width.** Black open circles: heterozygous controls; red closed circles: $drd^1$ homozygotes. (A) Plot of VM intensity measurements as a function of oocyte size ratio (A-P length/lateral width). There is no significant linear correlation between VM intensity and size ratio for either heterozygotes (p = 0.13) or homozygotes (p = 0.19, F test). (B) Comparison of VM intensity measurements between heterozygotes and homozygotes. Horizontal bars represent the means. The p-value is from a Mann-Whitney test comparing the two genotypes. (C) Plot of VM width measurements as a function of oocyte size ratio (A-P length/lateral width). There is no significant linear correlation between VM width and size ratio for either heterozygotes (p = 0.40) or homozygotes (p = 0.13, F test). (D) Comparison of VM width measurements between heterozygotes and homozygotes. Horizontal bars represent the means. The p-value is from a t-test comparing the two genotypes. n = 21 heterozygotes and 22 homozygotes.
(PDF)

**S1 Raw images. Images of Western blots used in Figs 3, 4, and 7.**
(PDF)

## Acknowledgments

We thank Dr. Celeste Berg for the CY2-GAL4 stock, Drs. Gail Waring and Anita Manogaran for antibodies and helpful discussions, and Anika Benske for technical assistance. Stocks obtained from the Bloomington Drosophila Stock Center (NIH P40OD018537) and the Vienna Drosophila Resource Center were used in this study.

## Author Contributions

**Conceptualization:** Edward M. Blumenthal.

**Formal analysis:** Tayler D. Sheahan, Amanpreet Grewal, Laura E. Korthauer, Edward M. Blumenthal.

**Funding acquisition:** Edward M. Blumenthal.

**Investigation:** Tayler D. Sheahan, Amanpreet Grewal, Laura E. Korthauer, Edward M. Blumenthal.

**Visualization:** Tayler D. Sheahan, Edward M. Blumenthal.

**Writing – original draft:** Edward M. Blumenthal.

**Writing – review & editing:** Tayler D. Sheahan.

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
