## [Decision Letter · Decision Letter 0]

5 Jun 2023

PONE-D-23-14466The *Drosophila drop-dead* gene is required for eggshell integrityPLOS ONE

Dear Dr. Blumenthal,

Thank you for submitting your manuscript to PLOS ONE. After careful consideration, we feel that it has merit but does not fully meet PLOS ONE’s publication criteria as it currently stands. Therefore, we invite you to submit a revised version of the manuscript that addresses the points raised during the review process.

We look forward to receiving your revised manuscript.

Kind regards,

Shubha Govind, PhD

Academic Editor

PLOS ONE

Journal Requirements:

Additional Editor Comments:

The Reviewers found this manuscript to contain novel findings, overall interesting, and pertinent to the field. Their reviews are detailed, with excellent feedback on experimental design, methods (including statistics), and data presentation. Addressing them all clearly will help improve the quality of this manuscript considerably.

Reviewers' comments:

Reviewer's Responses to Questions

**Comments to the Author**

1. Is the manuscript technically sound, and do the data support the conclusions?

Reviewer #1: Partly

Reviewer #2: Yes

Reviewer #3: Yes

2. Has the statistical analysis been performed appropriately and rigorously? 

Reviewer #1: I Don't Know

Reviewer #2: Yes

Reviewer #3: Yes

3. Have the authors made all data underlying the findings in their manuscript fully available?

Reviewer #1: Yes

Reviewer #2: Yes

Reviewer #3: Yes

4. Is the manuscript presented in an intelligible fashion and written in standard English?

Reviewer #1: Yes

Reviewer #2: Yes

Reviewer #3: Yes

5. Review Comments to the Author

Reviewer #1: This study, “Drosophila drop-dead gene is required for eggshell integrity” by Sheahan and colleagues, investigates the previously observed reduction in egg-laying of drop-dead (drd) mutant females. This phenotype was previously hypothesized to be due to starvation of drd mutant females because of defects in the gut epithelium. Upon feeding with a more nutritious diet (wet yeast), the drd females laid more eggs, but the researchers observed defects in the eggshell integrity of those eggs, as well as a reduced hatching rate.

The experiments performed by Sheahan et al., are basic, but sufficient for their story. In addition, the results reported will be of interest to both the Drosophila oogenesis community of scientists as well as those interested in the function of drd, which is yet unknown. Although the results included in this study does not elucidate the molecular role of drd in epithelia, the authors provide a better understanding of this gene’s role in Drosophila tissues. To be a more solid study, however, there are important controls that are missing throughout the study. These and other comments are outlined below.

Major comments:

1. The Introduction is missing a more complete explanation of the biology of the somatic follicle cells. The authors cite several helpful and comprehensive reviews, but a brief description of the follicle cells (beyond what is included in the first paragraph) will better provide readers of a broader audience the foundation of the organization and life cycle of the follicle cells. For example, it seems important to mention that the follicle cells form a polarized, mono-layered epithelium that lines the germline, with the apical domain facing the germline. This is relevant because it is at this interface that the vitelline membrane and chorion proteins are secreted and form, and that the follicle cells undergo cell death in stage 14 egg chambers.

2. In the Methods, the authors do not explain some of the scoring determinants and/or assays that were used throughout the study (nor do they explain this in the Results text). See major comments #3 and 4 below. Both analyses need to be better described in the Methods (the reference provided is not available on PubMed or Google Scholar).

3. Specifically, how were eggs determined to be fertilized or not? One widely used approach to do this is to mate drd mutant and knockdown females with a transgenic line (available from BDSC) expressing don juan fused to GFP (dj-GFP). In this line, the sperm tails are easily visible in the egg under a fluorescent dissecting microscope to confirm that fertilization has occurred.

4. Similarly, it is not clear how the authors determined when embryos arrested. They made the distinction of categorizing pre-gastrula vs. post-gastrula (although how they did this was not stated), but this is a very broad distinction. The data presented shows that 99% of embryos produced from drd mutant females arrest prior to gastrulation. These data suggest that there is a maternal-effect on embryogenesis; this observation will be of interest to the developmental biology community, although an additional experiment is necessary to better clarify the role of drd prior to zygotic genome activation just prior to gastrulation. It would be of great interest to determine in which cell cycle(s) the embryos arrest. Embryos collected from drd mutant and/or RNAi knockdown females can be collected (0-3h post egg-laying), aged for ∼3 hours, fixed, and stained for DNA (microtubule staining is helpful as well) to characterize the stage in which the embryos arrest. A presumed maternal-effect lethality would indicate the importance of drd in the syncytial blastoderm in which zygotic genome activation is not yet activated. If the authors are able to make this conclusion based on results that support this hypothesis, this would greatly improve the impact and interest of this work. This could then bolster the claim made in the Discussion (lines 532-533) regarding the developmental arrest hypothesized to occur in these embryos.

5. It was confusing as to why the authors used the drd1 allele solely and did not use the similarly severe allele drdlwf. It would be helpful confirmation to show that the phenotypes reported are consistent across both severe drd alleles, especially since they also included RNAi data to confirm the specificity of the phenotypes reported. In the Discussion (line 567), the authors stat that the drd1 mutant is phenotypically identical to drdlwf, but these experiments were not shown (and perhaps not performed) for the phentoypes assessed in this study. This claim is overstated if those experiments are not included.

6. There are many different statistical tests used throughout this study and they are stated in the text and figure legends but are not explained in the Methods. These should be better explained in the Methods section to clarify for readers why they used different statistical tests for different analyses.

7. The authors evaluated the timing of drd expression in oogenesis but did not show stages prior to drd expression (i.e., previtellogenic stages prior to stage 10). These data should be included to support the expression data referenced from Flybase. Did they indeed observe that Drd is not observed prior to stage 10B? Also, in order for someone to agree with the stages presented in Figure 1, the authors should include DNA staining merged and in separate channels. The GFP visualization alone is not clear enough to determine staging.

8. In Figure 1, it is difficult to discern the outlines of the egg chambers because only the GFP channel is shown. In addition to my suggestion above to include DNA staining, it would be very helpful if the authors presented the egg chambers as individual channels with callout letters instead of all in the same panel. Adding light microscope images or an outline of the egg chambers would be necessary (especially if DNA staining is not included as suggested).

9. This is also true for Figure S3 in which panel B shows an empty field. This is not meaningful if DNA staining or a light microscope image of what is present in the field of view is not included.

10. Is Drd present in ALL follicle cell (FCs) populations? For example, is Drd-GFP detected in centripetal cells, stretch cells, and in the epithelial monolayer, or is only present in a subset of FCs? Drd-GFP is present in the monolayer on the surface of the egg chamber, as shown in Figure 1, but the focal plane presented in Figure 1 only shows the surface of the egg chamber. Perhaps stretch cells also contain GFP but it’s difficult to say from the stage 10 egg chambers in Figure 1 what the GFP positive cells at the anterior are. Egg chambers shown as a longitudinal section through the middle of the egg chamber would help to elucidate the subpopulations of follicle cells in which Drd-GFP is present. This would be important to bolster their claims in the Discussion (paragraph starting on line 534).

11. None of the Western blots presented in the paper include loading controls. It is very difficult to fully believe the conclusions made, especially regarding the lack of protein detected in laid eggs, if a loading control is not included. I do not doubt that the results presented are real, but if these experiments are not well-controlled by showing that the same amount of protein is loaded per lane, the authors cannot make the conclusions presented for those figures.

12. Line 358: The authors remark that the decreased signal observed in Figure 4A for eggs laid was not consistent across replicates, but this was very striking. Why did the authors present data that was not representative? The decrease was so striking that I was going to suggest that the authors quantify the data but was surprised that they stated inconsistency of this decrease. This suggested a greater necessity to quantify these data. Quantification could be completed using a digital imager and software, or even simply stating how many replicates of the experiment showed the decrease would be perhaps sufficient.

13. In the data presented in Figure 5C, it was not clear to me if the egg chambers measured were egg chambers (prior to stage 14) or mature oocytes (stage 14). The authors use “egg chambers” and “oocytes” interchangeably in the text. This needs to be better clarified.

14. The authors used two follicle cell drivers for their RNAi knockdown experiments: CY-GAL4 and T155-GAL4. Both drivers are expressed in the follicle cells around stage 10, although perhaps at different levels, resulting in varying phenotypic difference. While the varying expressing has been reported before for those drivers, it would be very helpful if the authors also used a follicle cell driver that expresses earlier in oogenesis, such as traffic jam-GAL4 (tj-GAL4) to assess knockdown phenotype, which might be more robust under those conditions.

15. It would be extremely helpful for readers if the authors labeled the graphs in Figure 6 differently so that they’re easier to read. The genotypes are very long and for non-geneticists, this will be letter soup. Adding horizontal lines to the top of the graphs (as in the Western blot shown in Figure 7) would greatly help a reader visualize which genotypes are which. For example, a horizontal bar over the first 4 bars would indicate T155-GAL4 and then under that, 2 shorter horizontal lines above the pairs of bars (1 and 2 and then 3 and 4) can be used to indicate the RNAi lines used. Then in the axes for the x-axis, the authors can add “control” or “drd RNAi knockdown” as appropriate.

Minor comments:

1. It would be interesting to the field if the authors could comment about the formation of other important structures in the mature egg that form during oogenesis and may be impacted by proper deposition/development of the eggshell. Do the micropyles and dorsal appendages form properly? Experimental results showing and quantifying eggshell patterning defects might shed light on the precise defects that are found in eggs produced by drd mutant or knockdown females. In addition, results from these experiments might explain some of the phenotypes addressed earlier, such as the reported reduction of fertilized eggs produced by drd mutant or knockdown females.

2. The authors should add primary antibody dilutions used for Western blot analysis to the Methods.

3. The authors should include the camera used for confocal microscopy to the Methods.

4. Line 197: There is a grammatical error in this line (eclosion “and” placed on yeast paste).

5. It would have been better if the authors had used drd/+ controls for their experiments, so if the authors will be repeating experiments, it is suggested that they include this instead of drd/balancer.

6. In lines 262-264, the authors describe (albeit very briefly) how the data presented in Table 1 was collected. This information should be in the Methods, not the Results. Also, see my comment above about the need for more precise and informative methods regarding these assays.

7. In the Figures containing graphs, the authors follow a color scheme for the genotypes shown, but the figure legends do not provide a key for the colors used.

8. In the Figures containing graphs, it would be helpful if the authors include the precise “n” for each bar. The figure legends report a range of the number of samples included, but it would be more precise to include the exact “n” for each.

9. I was curious as to why the authors used life span to determine that progeny from females in which germline clones were induced were of the expected genotype. This seems like a good secondary confirmation, but simply sequencing the progeny would be more precise and a better primary confirmation of genotype. Even if the offspring are heterozygous, double peaks should be apparent by sequencing to confirm inheritance of maternal drd1.

10. The authors mention the number of eggs or egg chambers loaded per lane for the Western blots presented in their data; this information should be included in the Methods and not just in the associated figure legends.

11. Figure legend 4 (lines 362-365): This sentence read awkwardly and should be clarified.

12. Figure 5C: Statistical information is missing from the figure legend.

13. Throughout the description of the drd RNAi knockdown phenotypes, the authors simply refer to the drd knockdown females as “knockdown females”. It would be better if the authors add “drd” before this phrase to be more precise.

14. Line 419: the authors refer to one combination of driver and RNAi transgene to produce a specific effect, but they don’t state that in the text.

15. Table 2 should be reordered to show the control genotypes/phenotypes at the top and the drd RNAi knockdown genotypes/phenotypes in the 2nd row.

16. There is a concern in the fly community that UAS-Dcr2 presents phenotypes on its own during oogenesis. It would be helpful if the authors used a control that expressed UAS-Dcr2 under GAL4 control instead of GAL4-only controls.

17. When describing Figure 6 in the text, the authors state that there is a small, but significant increase in the fraction of eggs from drd knockdown females (lines 441-444), but this was only observed for specific combinations of GAL4 and UAS-RNAi lines. This should be better clarified in the text.

18. When describing the results of Figure 7, the authors report the number of biological replicates that showed the reported results. It was not clear if the authors were reporting that 3 of 3 replicates of GD3367 and 2 of 2 GD15915 displayed the phenhotye (lines 460-461). This should be clarified. Similarly, it would be a bit easier to follow if the authors phrased the data at the end of the same paragraph as 2 out of 4 (instead of 2/4) and 1 out of 3 (instead of 1/3).

19. In the legend for Figure 8, the authors should state that both drd mutant and control siblings were analyzed in the assay shown (and include those genotypes).

20. The authors are inconsistent in some of their uses of hyphenation or not throughout the text and figures. For example, sometimes “pre-gastrulation” or “post-gastrulation” are used, and other times both appear without a hyphen. This is also true for “hypo-osmotic”. It is suggested that the authors remain consistent in these terms.

21. The last sentence of the first paragraph on page 26 (lines 560-562) is confusingly written. Perhaps the authors mean to say that the study of Drd could prove to be useful for identifying additional factors needed for eggshell maturation? This should be clarified.

22. The authors should add callout letters to Figure 4.

23. In Figure 4, I was confused why 3h and 6h were so close together for the control lanes at the end of panel A. It seems the gel was cropped too closely and the spacing is not the same as for the mutant lanes.

24. In Figure 6, CY2-GAL4 should be consistently capitalized. In the 5th x-axis label for all panels (A-C), CY2-GAL4 is written as “Cy2-GAL4”.

25. Line 423: Change to “CY2-GAL4” (instead of CY-GAL4).

Reviewer #2: Sheahan et al.

The well-written manuscript presents important insight into the function of Dropdead in Drosophila. Loss of Dropdead is known to cause female sterility, but where it is required for fertility and how it impacts fertility was unknown. Taking advantage of the robust genetic tools of Drosophila, in combination with biochemical assessments, the study uncovers that Dropdead is required in the somatic follicle cells during mid to late oogenesis. There Dropdead is required for the proper crosslinking of the vitelline membrane, a key component of the eggshell. The study is well controlled and most phenotypes and changes are quantitatively assessed. A few minor edits will improve the paper, making it easier for a non-expert to fully appreciate the findings.

Minor issues:

1. More details are needed in some of the methods:

a. Visualization of drd expression – how are the egg chambers mounted?

b. Composition of 5X SDS-PAGE loading buffer

c. What primaries are used in the western blots and at what concentrations? I understand this is in the figure legends, but it should also be in the methods.

d. In immunostaining – how were the intensity measurements normalized?

2. Please add data mentioned but not shown.

a. RT-PCR (line 246)

b. Vm26Ab intensity data should be provided as graphs

3. The results section on the germline clones needs to be expanded to fully explain the clonal system being used. Many Drosophilists think of germline clones as making mosaic ovaries, but you are using the ovoD system. You need to explain how the system works and why the non-heat shock controls are sterile. It will be important to add some clarity on this method to the discussion as well

4. Expansion of drd-GAL4 GFP expression data – it would be very helpful for whole ovarioles to be shown (or more stages) to illustrate when it is not present and when it is. Also it appears that only some of the stretch follicle cells express it – is this true?

5. Add n’s to all graphs that don’t have them (Fig. 2, Fig 6

6. There are no loading controls for the western blots. Please either provide them or better explain why they are not needed.

7. Figure 3 legend text does not state what antibody is used.

8. Line 347: As expected,…. – it is unclear what sample(s) this sentence is referring to.

9. The order of the data presented in Figure 6 does not match the order discussed in the results.

10. Figure 6 title does not cover everything presented in the figure.

11. Figure 8 has very small sample sizes. It would be more convincing with larger numbers.

12. Lines 483-484 – missing figure panel callout or data not shown?

13. Typos:

a. Line 197: eclosion on placed on yeast paste with sibling males for two days

b. Line 239: The mutation in the latter of these alleles not been molecularly

Reviewer #3: This is a well-written manuscript that investigates the physiological defects that underlie the female sterile phenotype of mutations in the dropdead (drd) gene of Drosophila melanogaster. The authors show that the vitelline membrane portion of the eggshell for eggs laid by drd mutant females, is dye-permeable after the eggs are laid. Normally, the vitelline membrane becomes dye impermeable upon egg activation, when proteins are cross-linked by a poorly understood process. Because the drd sequence suggests that it is an integral membrane protein, consistent with published localization data, the authors hypothesized that Drd protein acts on proteins of the vitelline membrane in a necessary step towards cross-linking. Selective permeability of the vitelline membrane is critical for successful embryogenesis, so that gas exchange can occur, but desication does not.

The authors use flp-FRT-mediated mosaic analysis to show that the critical cell type for Drd gene expression is the somatic follicle cells that produce the components of the eggshell. This result is supported by similar results from dsRNA expression in the follicle cells using in vivo RNAi with follicle cell-specific Gal4 drivers. They further showed that an outer chorion eggshell protein was cross-linked normally in eggs laid by Drd mutant females, indicating a specific defect in vitelline membrane maturation.

To assess whether Drd was directly required during peroxidase-mediated glycosylation, the authors used an exogenous peroxidase method that should render dissected stage 14 immature eggs impermeable to chlorine bleach, even though they have not yet undergone egg activation. Although the control stage 14 immature eggs became impervious to bleach treatment, the immature eggs from stage 14 egg chambers did not.

Based on these results, it appears that Drd is necessary to make vitelline proteins available as substrates for the cross-linking reaction. This is an interesting finding, that will be useful to understanding this complex and poorly understood mechanism to mature the vitelline membrane.

Comments:

For the most part, the experiments are well-chosen and well-described. Sample sizes are appropriate; appropriate controls are used; and the results are critically interpreted.

However, the Western blot experiments lack information about replication. The methods are well-described, but it is not clear whether technical and/or biological replicates were performed for each stage examined.

I did not notice any typos, but did not search for them.

6. PLOS authors have the option to publish the peer review history of their article (what does this mean?). If published, this will include your full peer review and any attached files.

Reviewer #1: No

Reviewer #2: No

Reviewer #3: No

---

## [Author Response · Author response to Decision Letter 0]

24 Jul 2023

See separate "response to reviewers" file for detailed responses to all reviewer comments.

---

## [Decision Letter · Decision Letter 1]

31 Jul 2023

PONE-D-23-14466R1The *Drosophila drop-dead* gene is required for eggshell integrityPLOS ONE

Dear Dr. Blumenthal,

Thank you for submitting your manuscript to PLOS ONE. After careful consideration, we feel that it has merit but does not fully meet PLOS ONE’s publication criteria as it currently stands. Therefore, we invite you to submit a revised version of the manuscript that addresses the points raised during the review process.

We look forward to receiving your revised manuscript.

Kind regards,

Shubha Govind, PhD

Academic Editor

PLOS ONE

Reviewers' comments:

Reviewer's Responses to Questions

**Comments to the Author**

1. If the authors have adequately addressed your comments raised in a previous round of review and you feel that this manuscript is now acceptable for publication, you may indicate that here to bypass the “Comments to the Author” section, enter your conflict of interest statement in the “Confidential to Editor” section, and submit your "Accept" recommendation.

Reviewer #1: (No Response)

Reviewer #2: (No Response)

2. Is the manuscript technically sound, and do the data support the conclusions?

Reviewer #1: Partly

Reviewer #2: Yes

3. Has the statistical analysis been performed appropriately and rigorously? 

Reviewer #1: Yes

Reviewer #2: Yes

4. Have the authors made all data underlying the findings in their manuscript fully available?

Reviewer #1: Yes

Reviewer #2: Yes

5. Is the manuscript presented in an intelligible fashion and written in standard English?

Reviewer #1: Yes

Reviewer #2: Yes

6. Review Comments to the Author

Reviewer #1: The authors addressed most of the textual changes suggested by the reviewers and the manuscript is stronger because of these changes. However, there are important experimental controls that are still missing because the authors did not adequately address the previous reviews. These controls are essential to support the conclusions made and should be included to uphold scientific expectations of well-controlled experimentation.

Major comments:

1. In their response to reviewers and in the manuscript, the authors reference Elalayli et al. (2008) as an example of a study characterizing the mutant phenotype of a vitelline membrane (VM) gene (palisade) with aberrant VM structure. Elalayli et al. perform additional experiments to characterize the mutant phenotype; these experiments were not included in the study by Sheahan et al. despite reviewer requests for similar experiments. For example, confirmation of fertilization using a tagged sperm tail line and evaluation of the stages at which embryonic arrest occurs were shown by Elalayli et al. and suggested by the reviewers, but not included in the revised manuscript. In the revised manuscript, Sheahan et al. use yolk composition to determine fertilization status. I am inclined to accept this less precise method but believe that the lack of staging for the observed arrest is problematic, since the authors’ main conclusions impinge on the timing at which Drd functions in VM formation.

a. Mutant alleles of genes involved in VM composition, secretion, and cross-linking typically lay unfertilized eggs due to defects in eggshell morphology or integrity (resulting in gas exchange and micropyle defects) or cell-cell signaling required for proper dorsal-ventral patterning. Some eggs derived from mutant females, however, are fertilized and those embryos typically arrest in gastrulation (again, likely due to patterning defects). drd-derived embryos may have distinct phenotypes from typical VM gene alleles, but these differences should be better characterized and comparisons should be made to place Drd in specific roles in VM cross-linking, as proposed by the authors.

b. The authors characterized the observed sterility in mutant- and RNAi-derived embryos (Tables 1 and 2) but should perform more precise staging of these embryos to clarify if their arrest is due to processes distinct from VM-associated defects shown for other VM genes. The authors show that 99% of drd¬¬-derived embryos arrest prior to gastrulation, suggesting maternal and not zygotic effects. To my knowledge, syncytial blastoderm arrest (i.e., prior to cellularization) is not usually associated with mutants for genes involved in VM cross-linking. Evidence that cellularization does not occur in drd-derived embryos may suggest another role for Drd in embryogenesis, thereby highlighting the importance of this assay. The results could simply be grouped by pre-cellularization vs. post-cellularization (instead of exact cell cycle stages) among pre-gastrulation arrested embryos. If the authors already have pictures of DAPI-stained embryos, fine-tuned staging quantitation should be straight-forward to perform.

c. Have the authors considered that Drd has two functions – one in VM cross-linking (although the authors suggest that Drd’s effect on VM is likely indirect) and the other in early embryonic signaling/cell cycles (which may be direct or indirect)? If Sheahan et al. quantified the arrest timing of drd1-derived embryos, these results would advance their understanding of Drd’s functions in vivo.

2. As pointed out by two reviewers, none of the Western blots presented in this study include loading controls. Although the authors responded as to why they thought this information would not be helpful, I disagree. The lack of protein detected cannot be claimed to be informative if it is not shown experimentally that protein was indeed loaded in those lanes. In the revised manuscript, the authors simply state that the same number of egg chambers or embryos were loaded per lane, but without experimental confirmation, the interpretation of their results, which is vital to the main conclusions of this study, is weakly supported experimentally. Identically loaded Coomassie-stained gels or Ponceau S-stained membranes before blotting would serve as useful loading controls (included in the supplement) if the authors were not able to use a typical loading control for technical reasons.

3. In the revised manuscript, the authors include a more comprehensive explanation of the ovoD system used to assess the role of drd in the germline vs. somatic cells. This addition is very helpful for the readers. The authors then conclude that drd is only required in somatic cells because 15/17 females were fertile after heat shock treatment. How did the authors assess fertility for this experiment? This information should be added to the Methods. In addition, these results seem contradictory to other results shown in the study wherein larval hatching is severely disrupted (Tables 1 and 2). If the authors used egg-hatching to confirm fertility, then these data do not support their claim that drd-derived embryos exhibit maternal-effect lethality. A better explanation of these contradictory results is requested.

Minor comments:

1. As mentioned previously, UAS-Dcr2 presents phenotypes on its own during oogenesis. The authors would not need to redo every experiment to include this control, as they suggest. Instead, simply collecting and analyzing UAS-Dcr2/+; GAL4/+ (for example) embryos under the same conditions used in the RNAi experiments could be done and compared to the RNAi knockdown results.

2. In the revised text, the authors refer to Figure S1 stating that RT-PCR was used to confirm that a shortened drd transcript was produced in drd1 mutants (lines 291-292), but it is not entirely clear how RT-PCR data was included in Fig. S1. I think the authors simply mean that RT-PCR was used to synthesize cDNA that was used for Sanger sequencing, but this is not clear from the text. Either RT-PCR results should be shown (as bands or in a normalized graph), or the Results and Methods should be clarified since the drd1 allele is used throughout their study.

Reviewer #2: The majority of the issues have been addressed. However, I believe the authors' did not upload the revised SFig 1. with the RT-PCR data. Further, the methods are still unclear for how intensity measurements were made (details/program/normalization or not needed). Finally, while the n's are added to Fig 2 and 6 legends, to be consistent with other figures they should be shown in the actual figures.

7. PLOS authors have the option to publish the peer review history of their article (what does this mean?). If published, this will include your full peer review and any attached files.

Reviewer #1: No

Reviewer #2: No

---

## [Author Response · Author response to Decision Letter 1]

12 Oct 2023

Please see "response to reviewers" file

---

## [Editor Report · Decision Letter 2]

14 Nov 2023

PONE-D-23-14466R2The *Drosophila drop-dead* gene is required for eggshell integrityPLOS ONE

Dear Dr. Blumenthal,

Thank you for submitting your manuscript to PLOS ONE. After careful consideration, we feel that it has merit but does not fully meet PLOS ONE’s publication criteria as it currently stands. Therefore, we invite you to submit a revised version of the manuscript that addresses the points raised during the review process.

There is a slight difference in the way the Reviewers and I have been looking at the Western images included in the manuscript from the way you want us to look at them. I think we get that the amounts of proteins loaded in the lanes is about equal as you controlled for the starting material that went into the protein preparation protocol.

We also understand that you are not making a quantitative argument; you are using the Western results to highlight drd's function in egg shell assembly/VM protein cross-linking, a process, that is temporally controlled.

We also see that your results for both Cp36 and Vm26Ab are consistent with previous reports.

Our focus now is on the lanes in all the blots where there is no signal. What we are looking for is evidence, that in these lanes, protein was loaded. This is important, because the negative results in the lanes in question are helping you build your story. While doing so, it is important to be convinced that the lanes in question actually received protein extract. This is standard practice these days and a valid request from the Reviewers.

The most straightforward solution would be for you to re-probe the blots shown in the manuscript with anti-lamin, anti-actin, or anti-tubulin (or some other oogenesis-related) antibody to show that the lanes without the signal contain protein. This will help win over the most skeptic of us about the validity of your results. I understand that this is a small point in the larger context of your manuscript, but I believe that not having this control will leave open the discussion whether protein sample was/was not loaded in the lanes without signal.

An additional solution would be for you to review your existing trial and final films from overexposed blots in which the background signal in the lanes without the signal matches the background signal in the lanes with signal. It would be best however to include images of such films (or ones from re-probed blots) in the supplement to explicitly make this point.

In doing so, please organize the data from the additional films into a supplemental figure where the relationship of the newly presented films (and lanes) with that of the ones already included in the manuscript is clearly stated to specifically address this point.

We look forward to receiving your revised manuscript.

Kind regards,

Shubha Govind, PhD

Academic Editor

PLOS ONE

---

## [Editor Report · Decision Letter 3]

22 Nov 2023

The *Drosophila drop-dead* gene is required for eggshell integrity

PONE-D-23-14466R3

Dear Dr. Blumenthal,

We’re pleased to inform you that your manuscript has been judged scientifically suitable for publication and will be formally accepted for publication once it meets all outstanding technical requirements.

Kind regards,

Shubha Govind, PhD

Academic Editor

PLOS ONE
---

## [Editor Report · Acceptance letter]

27 Nov 2023

PONE-D-23-14466R3 

The *Drosophila drop-dead* gene is required for eggshell integrity 

Dear Dr. Blumenthal:

I'm pleased to inform you that your manuscript has been deemed suitable for publication in PLOS ONE. Congratulations! Your manuscript is now with our production department. 

Kind regards, 

on behalf of

Dr. Shubha Govind 

Academic Editor

PLOS ONE